

# Nitrogen fertilizer application for improving the biomass, quality, and nitrogen fixation of alfalfa (*Medicago sativa* L.) at different growth stages in a saline–alkali soil

Weifan Wan[1,2], Qian Liu[1], Ke Li[1], Kun Zhao[1], Fei Qi[1], Yuanshuo Li[1], Zhi Sun[1] and Haigang Li[1]

[1] Inner Mongolia Key Laboratory of Soil Quality and Nutrient Resources, Key Laboratory of Agricultural Ecological Security and Green Development at Universities of Inner Mongolia Autonomous Region, Inner Mongolia Agricultural University, Hohhot, China
[2] Inner Mongolia Academy of Science and Technology, Hohhot, Inner Mongolia, China

Corresponding author
Haigang Li, haiganglis@imau.edu.cn

## ABSTRACT

**Background:** The application of nitrogen (N) fertilizer to alfalfa (*Medicago sativa* L.) has received little attention due to the ability of this plant to fix N. However, N deficiency stress is often observed in marginal lands of China, especially in saline–alkali soils. Thus, this study aimed to assess the response of alfalfa yield, quality, N fixation, and soil N concentration to N fertilizer application at different stages in saline–alkali soil of Inner Mongolia. A 2-year (2020 and 2021) field experiment with five N fertilizer application rates, namely, 0 (N0), 20 (N20), 60 (N60), 120 (N120) and 180 (N180) kg N ha$^{-1}$, was conducted in Inner Mongolia.
**Results:** The results showed that N fertilizer application of 180 kg ha$^{-1}$ significantly increased the total alfalfa yield by 29%–32% by improving the stem–leaf ratio; however, it didn't lead to a further increase in alfalfa quality. N fertilizer applications of 60 and 120 kg ha$^{-1}$ significantly improved the crude protein by 10.6%–22.7% and reduced the acid or neutral detergent fiber by 10.0%–18.7% in vegetative and bud stages, respectively, by improving the leaf N concentration. Furthermore, the fraction of N derived from the atmosphere reached 68.6%, with a significant increase in the corresponding amount of N fixed in N60. Soil NO$_3^-$-N concentrations significantly increased by 24.1%–33.3%, and NH$_4^+$-N concentrations increased by 1 to 3 times when N fertilizer application exceeded 120 kg N ha$^{-1}$ compared with that in N0.
**Conclusions:** Overall, this study revealed the essential role of N fertilizer application at low rates in alfalfa production, as this practice not only increases alfalfa yield but also improves N fixation in saline–alkali soil. However, it did not result in further improvement in alfalfa quality at the early flowering stage. The findings provides valuable guidance for N fertilizer application in alfalfa production on saline–alkali soils.

## INTRODUCTION

Alfalfa (*Medicago sativa* L.) is an important legume forage grass with high yield, good palatability, and high nutritive value and is widely cultivated in China (*Wang & Zou, 2020*). According to *National Livestock Stations (2015)*, China's production of alfalfa hay amounted to 32.17 million tons, with only 1.8 million tons meeting the criteria for high quality. Thus, alfalfa production cannot meet the demands of recent increases in animal husbandry, and 49% of the alfalfa required for China is imported from the USA and other regions (*Wang, 2017*; *Wang & Zou, 2020*). The positive correlation of growth and biomass with the availability of soil nutrients has been widely recognized in forage crops (*Xie et al., 2015*; *Chippano et al., 2020*). To ensure food security in China, alfalfa is often cultivated on marginal lands, which typically have poor soil fertility and suboptimal climatic conditions (*Yang et al., 2013*). Approximately 25% of China's farmland is salt-affected, mainly located in arid and semi-arid regions (*Zhao & Li, 1999*). Salinity is a crucial factor that limits both the yield and quality of alfalfa (*Peel et al., 2004*; *Nadeem et al., 2019*). Thus, fertilizer application is an essential measure for achieving high yields and producing high-quality alfalfa in a saline–alkali soil.

Numerous studies over several decades have examined the impact of mineral fertilizers on alfalfa, and these studies have consistently shown that phosphorus (P) and potassium (K) fertilizer application significantly improves the yield and quality of alfalfa (*Berg et al., 2005*; *Gu et al., 2018*). Alfalfa, like other legumes, forms a symbiotic relationship with rhizobia bacteria to fix atmospheric nitrogen (N), reducing its dependence on external N inputs while enriching soil fertility for subsequent crops (*Yost, Russelle & Coulter, 2014*; *Riedell, 2014*). Thus, N fertilizer has received less attention from alfalfa plants due to its ability to fix N. However, stress due to N deficiency is often observed on marginal lands and inhibits forage growth (*He & Li, 2017*). The N application is particularly important in saline–alkali soils because these conditions hinder rhizobial activity and N fixation, leading to a need for external N inputs (*Elgharably & Benes, 2021*). *Kamran et al. (2022)* reported that N application increases alfalfa yield at a rate of 150 kg N ha$^{-1}$. In contrast, a 3-year field study showed no significant differences in alfalfa yield among different N fertilizer application rates (*He, Xie & Li, 2018*). The contradictory findings on the effects of fertilizer application on alfalfa yield and quality have created confusion in providing clear guidance for farmers in alfalfa production, while also limiting the application of UAV precision agriculture technologies (*Sharma et al., 2022*).

Crude protein (CP), neutral detergent fiber (NDF), and acid detergent fiber (ADF) directly reflect the nutritional quality of forages and are important indicators for evaluating alfalfa grade (*Chen et al., 2020*; *Hakl et al., 2021*). In addition, N application increases CP by increasing the number of leaves (*Slamet et al., 2012*). In contrast, a study showed that N fertilizer application usually leads to an increase in the nonprotein N content and a decrease in alfalfa CP (*Atanasova, 2008*). Moreover, excessive fertilizer application above plant demand increases the concentrations of ADF and NDF, which leads to a decline in alfalfa quality (*Fan et al., 2018*; *Jungers et al., 2019*). Excessive N fertilizer application can also have broader negative impacts, including increasing N leaching risk, contributing to

 

environmental problems such as eutrophication (*Galloway et al., 2008*). Additionally, high nitrate levels in forage can cause nitrate toxicity of livestock (*Kellems & David, 2002*). Thus, a further research is needed to clarify optimal N fertilizer rate in high-quality alfalfa production on marginal lands, and environmental safety in saline–alkali soils.

The stability and activity of nodulation decline in poor soil and suboptimal environmental conditions, such as salt-affected soils and dry and cool weather (*Jing et al., 2010*; *Libault, 2014*). The N fertilizer application effectively increases the amount of N fixed and N fixation under these conditions (*Elgharably & Benes, 2021*; *Wan et al., 2023*). However, excessive N fertilizer application reduces N fixation capacity (*Salvagiotti et al., 2008*; *Dai et al., 2018*; *Bahulikar et al., 2021*).

Alfalfa quality decreases with increasing biomass from emergence (regeneration) to harvest (*Putnam et al., 1999*; *Undersander, 2011*). However, the impacts of dynamic changes in alfalfa yield, N fixation, and the nutritive value of biomass caused by N fertilizer application at different growth stages are unknown. This study seeks to fill this gap in saline–alkali soils, offering new insights into optimizing N management for enhanced productivity and sustainability in these challenging environments. In this study, our objectives were to (i) determine the dynamics of alfalfa biomass and N accumulation under different N application rates, (ii) investigate the correlation between soil N supply and N fixation, and (iii) assess the contribution of N fertilizer application to alfalfa quality.

## MATERIALS AND METHODS

### Experimental site, soil, and climate

A 2-year field experiment was conducted beginning in 2020 at the Semiarid Ecosystem Research Station (40°38′N, 111°28′E, altitude 1,702 m) in the Tumd Left Banner, Hohhot County, Inner Mongolia, China. Meteorological data were presented in Fig. 1A. The site has a moderate temperate and semiarid climate, with an annual average air temperature of 6.7 °C, average monthly maximum temperature of 17.0–22.9 °C (July), and average monthly minimum temperature of −12.7 °C to 14.4 °C (January). The soil was classified as a light soda-alkalized aqueous soil according to Chinese soil taxonomy. Maize (*Zea mays* L.) was sown for 2 years before the experiment was established. Table 1 shows the soil properties at the start of this experiment.

### Experimental design

The field experiment was conducted for 2 years with five N application rates (in kg N ha$^{-1}$) applied every year—0 (N0), 20 (N20), 60 (N60), 120 (N120) and 180 (N180)—and was designed according to randomized complete block design. The N fertilizer treatments were designed based on previous meta-analysis results, which suggested that the optimal N fertilizer application rate ranges between 30 and 60 kg ha$^{-1}$ (*Wan, Li & Li, 2022*). The treatment schematics are shown in Fig. 1B. There were three replicates in each treatment. All the fertilizers were broadcast into each plot in the seeding year (2020) and applied to furrows close to the root systems of the alfalfa plants in 2021. The different fertilizer application method is due to alfalfa a perennial plant. The plot size was 60 m$^2$ (12 m × 5 m), and the plots were treated with the same N application rates in both 2020 and 2021.

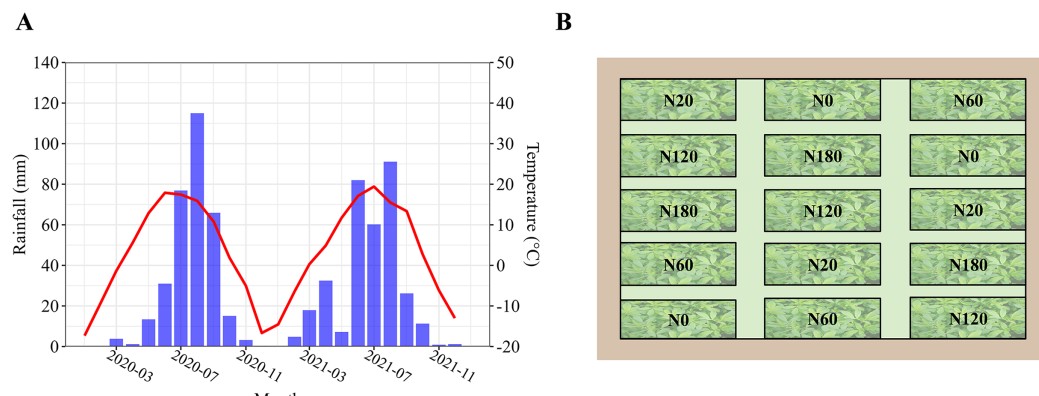

**Figure 1 Annual rainfall (blue bar) and mean temperature (red line) of 2 years (2020 and 2021) in research station (A), and schematic diagram of the experimental design for nitrogen application rates (B).**

**Table 1 Soil physical and chemical properties (0–30 cm) prior to the establishment of the experiment.** The data are presented as the means ± standard deviations (SDs).

| Variable | Unit | Mean ± SD |
|---|---|---|
| Soil pH | – | 8.68 ± 0.16 |
| Electrical conductivity | µS/cm | 190.15 ± 15.10 |
| Soil $NH_4^+$-N | mg/kg | 2.34 ± 0.40 |
| Soil $NO_3^-$-N | mg/kg | 12.61 ± 7.67 |
| Soil Olsen P | mg/kg | 15.63 ± 4.36 |
| $NH_4OAc$-K | mg/kg | 85.6 ± 8.66 |
| Soil organic matter | g kg$^{-1}$ | 7.18 ± 2.37 |
| $Ca^{2+}$ | mg/kg | 43.25 ± 2.45 |
| $Mg^{2+}$ | mg/kg | 47.56 ± 5.15 |
| $K^+$ | mg/kg | 21.32 ± 3.98 |
| $Na^+$ | mg/kg | 25.98 ± 5.33 |
| $HCO_3^-$ | mg/kg | 23.67 ± 3.76 |
| $Cl^-$ | mg/kg | 19.58 ± 5.22 |
| $SO_4^{2-}$ | mg/kg | 54.88 ± 9.88 |
| Soil sandy | % | 64.47 ± 3.71 |
| Soil silt | % | 29.56 ± 2.59 |
| Soil clay | % | 5.97 ± 1.48 |

Alfalfa (*Medicago sativa* L. cv. Zhongmu. No. 1) was sown on June 10, 2020, at a rate of 15 kg seed ha$^{-1}$. Every plot had a 20 cm spacing between rows and 50 cm of buffer space between adjacent plots. All the plots were treated with N fertilizer as urea (46% N), 120 kg $P_2O_5$ ha$^{-1}$ as calcium superphosphate (18% $P_2O_5$) and 150 kg $K_2O$ ha$^{-1}$ as $K_2SO_4$ (50% $K_2O$). Microsprinkler irrigation with 140 mm of water was applied on June 25 and August 16, 2020, and April 17, May 8, May 20, June 21, July 8 and September 10, 2021, respectively. Each plot was irrigated using a micro sprinkler system with a flow rate of

30 L/h. The irrigation was conducted for 6 hours per event. To ensure uniform water distribution across all plots, the system was calibrated prior to the experiment, and regular checks were conducted during irrigation to confirm consistency in water application. This ensured that alfalfa growth was not affected by water limitation. Weeds and pests were managed through the application of herbicides on July 3 and insecticides on August 19, 2020. No herbicides or insecticides were applied in 2021, as pest pressure was minimal and no significant weed issues were observed, allowing us to maintain the field without chemical interventions.

## Sampling and measurements

In 2020, alfalfa shoots were sampled on August 6 (vegetative stage), August 27 (vegetative stage), and October 3 (vegetative stage, cutting). In 2021, alfalfa shoots were sampled on May 23 (vegetative stage), June 2 (bud stage), June 15 (early flowering stage, the first cutting) at first growth, July 16 (vegetative stage), July 30 (bud stage), August 6 (early flowering stage, the second cutting) at second growth, and September 23 (vegetative stage, the third cutting) at third growth. In the seeding year (2020), alfalfa did not reach the bud stage before October due to insufficient temperature accumulation. At each sampling, plants in 50 cm sections of a row, excluding the edge rows, were randomly selected and cut from the ground. This sampling process was repeated five times in each plot. To minimize the impact of edge effects, all sample collections were conducted from the inner sections of each plot, deliberately avoiding the outer rows. The fresh weight of each forage sample was measured. Approximately 100 g of sample was separated into stems and leaves, which were weighed and oven-dried at 105 °C for 30 min (to remove enzymes) and then at 75 °C for 72 h (*Fan et al., 2016*). Finally, the dry weight/fresh weight ratio was calculated. The biomass of alfalfa (kg ha$^{-1}$) was calculated for the land area on a dry mass basis. And alfalfa nodule number was counted at each sampling. The biomass in each cutting was recorded as the yield, including that on October 3, 2020, and June 15, August 6 and September 23, 2021. The yield in 2021 was the total biomass of all the cuttings performed during that year.

All the oven-dried stem, leaf, and root samples were ground to powder for measurement of plant N concentrations, which were determined using the auto-Kjeldahl method after $H_2SO_4$-$H_2O_2$ digestion (*International Organization for Standardization, 1998*). The shoot N uptake (kg ha$^{-1}$) was calculated by multiplying the shoot N concentration by the forage yield.

Stem and leaf samples were analyzed for shoot $\delta^{15}N$ using an isotope facility (Iso-prime 100; Elementar, Ronkonkoma, NY, USA). The fraction of N derived from the atmosphere (%Ndfa) in the harvested alfalfa material was calculated as follows (*Fan et al., 2006*): % Ndfa = ($\delta^{15}N$ reference plant − $\delta^{15}N$ alfalfa/$\delta^{15}N$ reference plant − β) × 100, where the $\delta^{15}N$ reference plant is $\delta^{15}N$ value for a nonleguminous plant (*Echinochloa crus-galli* L. P. Beauv) under the same conditions and β is the $\delta^{15}N$ of alfalfa grown hydroponically in a phytotron, where the plant was entirely reliant on N fixation for its N nutrition. The amount of N fixed by the alfalfa plants was calculated as follows: N fixed (kg ha$^{-1}$) = %Ndfa (%) × shoot dry weight (kg ha$^{-1}$) × shoot N concentration (%).

The CP concentration was calculated as 16% of the N concentration (*Lourenco et al., 2002*). The NDF and ADF concentrations were determined by the Van Soest method (*Van Soest, Robertson & Lewis, 1991*). About 0.5 g of plant sample was weighed into the filter bag and sealed. The samples were measured using an ANKOM2000i automatic fiber analyzer. After the washing process was completed, the bags were rinsed with acetone, dried, and then weighed for final calculation.

At each plant sampling, ten soil subsamples were taken randomly from 0–20 to 20–40 cm soil layers in each plot and mixed to form composite soil samples. Nitrate-N and ammonium-N were measured using an autoflow injection system (Skalar, Breda, The Netherlands) after the soil samples (10 g) were soaked in 2 mol L$^{-1}$ KCl and shaken at 200 rpm for 1 h.

### Statistical analyses

All parameters were analyzed by SAS (SAS v8.0; SAS Institute Inc., Cary, NC, USA). The least significant difference (LSD) at $P = 0.05$ was considered significant. Prior to conducting the analysis of variance (ANOVA), the normality of the data was assessed using the Shapiro–Wilk test. The results indicated that all data were normally distributed ($P > 0.05$), meeting the assumptions required for ANOVA. The results of ANOVA for parameters among N fertilizer application rates and dates showed in Tables S1 and S2. Figures were generated using the ggplot, cowplot, rstatix, and car packages in RStudio software (version 4.2.1).

## RESULTS

### Alfalfa biomass and yield components

In 2020, alfalfa height, stem diameter, and branch number increased significantly over time. Those yield components did not significantly differ among the N fertilizer application rates (Figs. 2A, 2C, and 2E). The alfalfa height ranged from 20 to 61 cm, and the stem diameter ranged from 1.2 to 3.1 mm, which was lower than that in 2021. The range of branching number was the same between the 2 years. In 2021, the alfalfa height in N180 was 9.6%–11.1% greater than that in N0 and N20 on May 26 and June 3, respectively (Fig. 2B). Stem diameter in N180 was 20%–28%, which was significantly greater than that in N0, N20 and N60 on May 26 (Fig. 2D). The lowest branching number was observed in the treatments with no N application on June 3, June 15, and July 30. In addition, there was no difference in yield components (height, stem diameter or branching number) among the N application rates in 2021 (Fig. 2F).

Alfalfa biomass increased with increasing N fertilizer application in 2020 and 2021. In 2020, there was no difference in alfalfa biomass among the N application rates on August 6 and August 27 (Fig. 3A). The cutting biomass (yield) of alfalfa in N180 was 31%–32% greater than that in N0 and N20 in 2020 (Figs. 3A and 3C). In 2021, the alfalfa biomass significantly decreased with the number of cuttings. On May 26 and June 3 (the first regrowth), alfalfa biomass significantly increased by 30%–52% in N180 compared with that in N0 and N20. On June 15 after the first cutting, the alfalfa biomass in N180 was 29%, which was significantly greater than that in N0. On July 16 and July 30 (the second

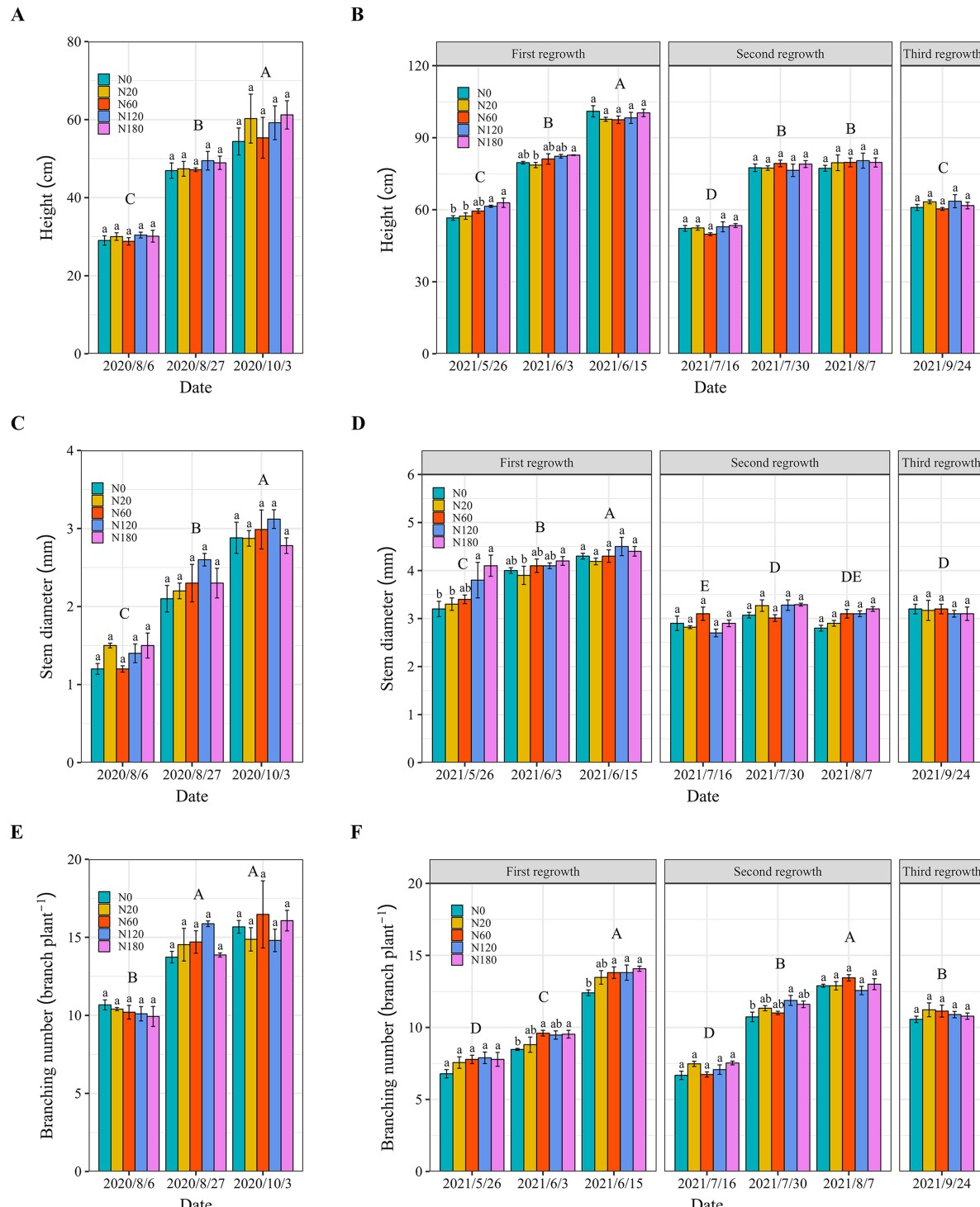

**Figure 2 Alfalfa height (A and B), stem diameter (C and D) and branching number (E and F) under different N application rates and on different dates in 2020 and 2021.** The data are presented as the means ± SEs (*n* = 3). The *P* values for the main interactive effects of N fertilizer application rates (N) and date are reported in Tables S1 and S2. Different capital letters indicate significant differences among the dates. The vertical bars with different lowercase letters associated with each date indicate significant differences among the treatment means. The five N fertilization treatments are 0 (N0), 20 (N20), 60 (N60), 120 (N120) and 180 (N180) kg N ha$^{-1}$. The dates in 2020 are both in the vegetative stage. In 2021, the vegetative stage is represented by 2021/5/26, 2021/7/16 and 2021/9/24; the bud stage is represented by 2021/6/3 and 2021/7/30; and the early flowering stage is represented by 2021/6/15 and 2021/8/7. The cutting dates were 2020/10/3, 2021/6/15, 2021/8/7 and 2021/9/24.

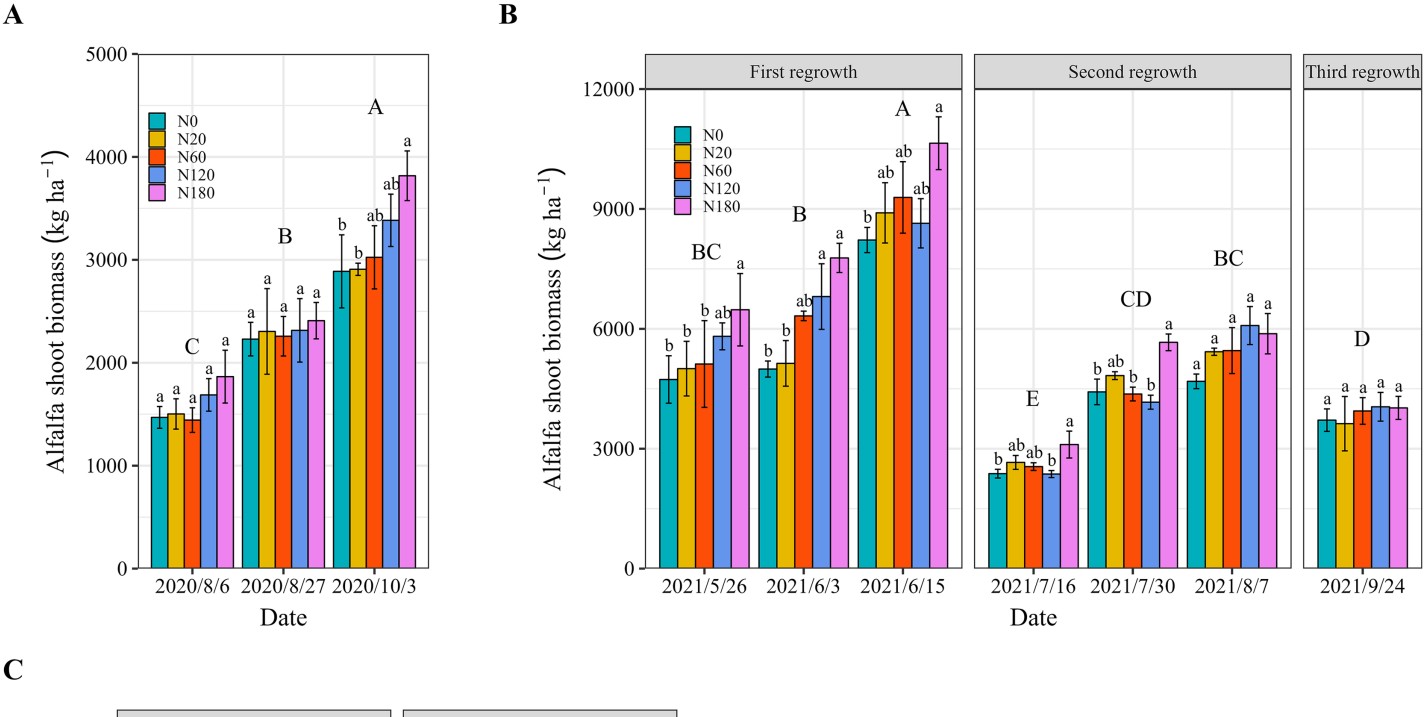

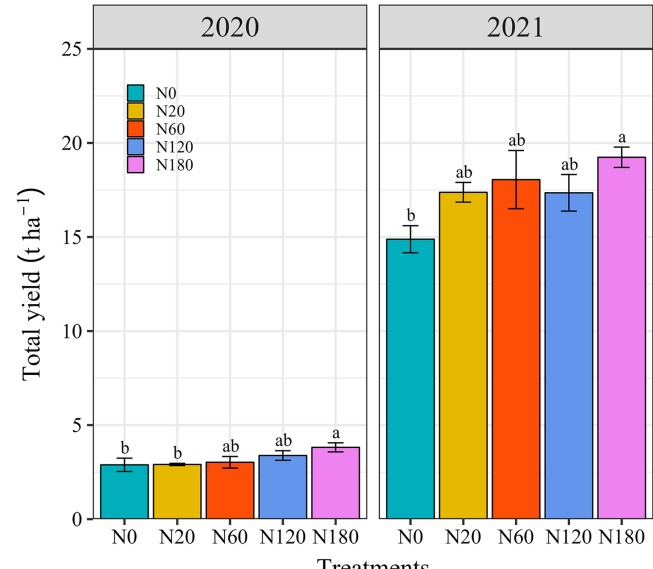

**Figure 3 Alfalfa biomass under different N application rates and on different dates in 2020 (A) and 2021 (B). (C) Shows the total yield in 2020 (2020/10/3) and 2021 (the sum of yield of 2021/0615, 2021/08/07 and 2021/09/24).** The data are presented as the means ± SEs (*n* = 3). The *P* values for the main interactive effects of N fertilizer application rates (N) and date are reported in Tables S1 and S2. Different capital letters indicate significant differences among the dates. The vertical bars with different lowercase letters associated with each date indicate significant differences among the treatment means. The treatments and dates are the same as those described in Fig. 2.   

regrowth), the alfalfa biomass in N180 was still significantly greater than that in N0. However, the 180 kg N ha$^{-1}$ fertilizer application did not lead to a further increase in alfalfa biomass at cutting for the second or third regrowth in 2021 (Fig. 3B). The total alfalfa yield was lower in 2020 than in 2021 (Fig. 3C). Overall, the application of 180 kg N ha$^{-1}$ of

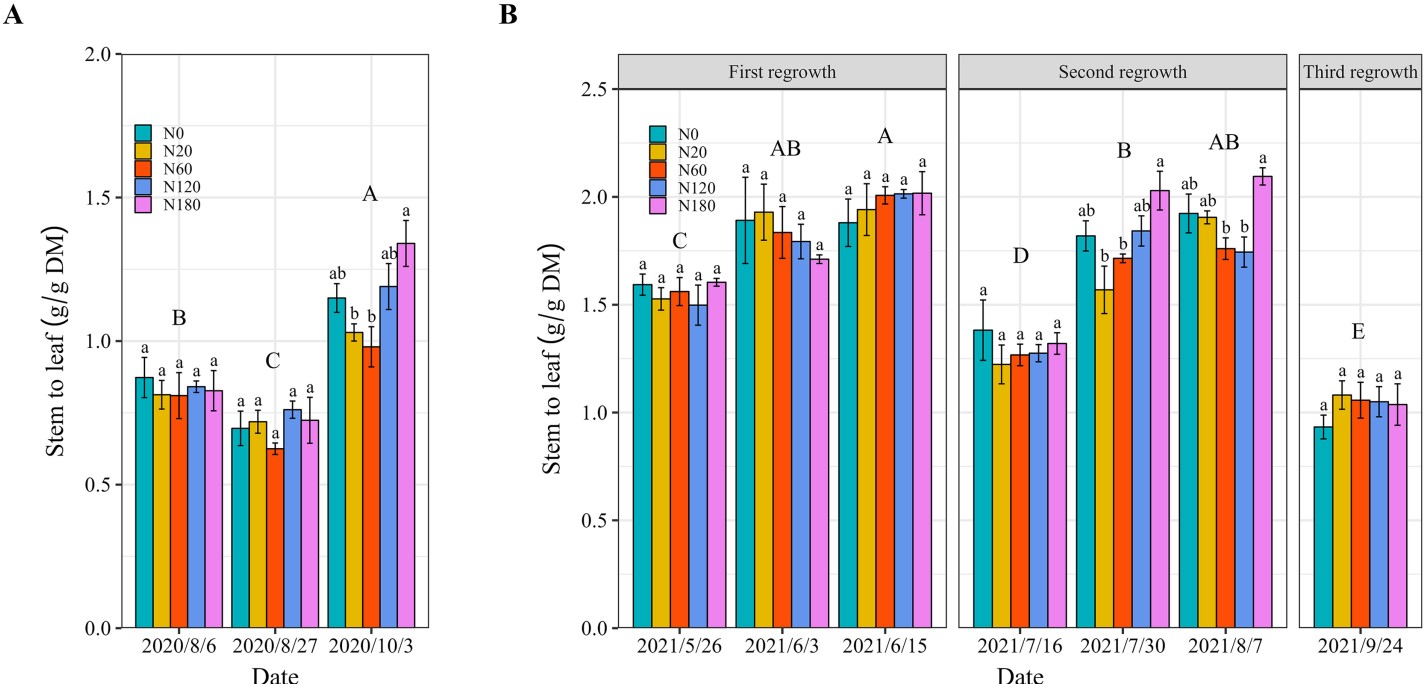

**Figure 4 Alfalfa stem-to-leaf ratio under different N application rates and on different dates in 2020 (A) and 2021 (B).** The data are presented as the means ± SEs (*n* = 3). The *P* values for the main interactive effects of N fertilizer application rates (N) and date are reported in Tables S1 and S2. Different capital letters indicate significant differences among the dates. The vertical bars with different lowercase letters associated with each date indicate significant differences among the treatment means. The treatments and dates are the same as those described in Fig. 2.

fertilizer increased the biomass and significantly improved the total yield by 29%–32% compared to N0.

Compared with N180, N application in N20 and N60 significantly decreased the stem-to-leaf ratio by 23% and 26%, respectively, at the cutting on October 3, 2020 (Fig. 4A). Similarly, N application at a rate of 60 kg N ha$^{-1}$ led to significant decreases in the stem-to-leaf ratio of 14.6% and 15.9% on July 30 and August 7 (second regrowth) 2021, respectively (Fig. 4B). The application of 60 kg N ha$^{-1}$ did not lead to a further decrease in the alfalfa stem-to-leaf ratio on other dates.

## Alfalfa leaf, stem, and root N concentrations and shoot N uptake

Alfalfa leaf concentration decreased significantly with alfalfa growth (regrowth). The N fertilizer application increased the leaf N concentration when the rate was greater than 20 kg N ha$^{-1}$. Similarly, the leaf N concentration in the N0 treatment was 7.8%–23.5% and 6.5%–29.2% lower than that in the N fertilizer application treatments in 2020 and 2021, respectively (Figs. 5A and 5B). In 2021, N application in N180 did not result in a significant increase in the leaf N concentration on May 26 or June 3. There was no significant difference in the leaf N concentration among the N fertilizer rates on June 15, August 7 and September 24. In 2020, the stem N concentration significantly increased (21.9%–38.3%) only when the N fertilizer application rate exceeded 60 kg N ha$^{-1}$ on August 8 (Fig. 5C). In addition, it declined in N180 on June 3 and August 7, 2021 (Fig. 5D). There was no

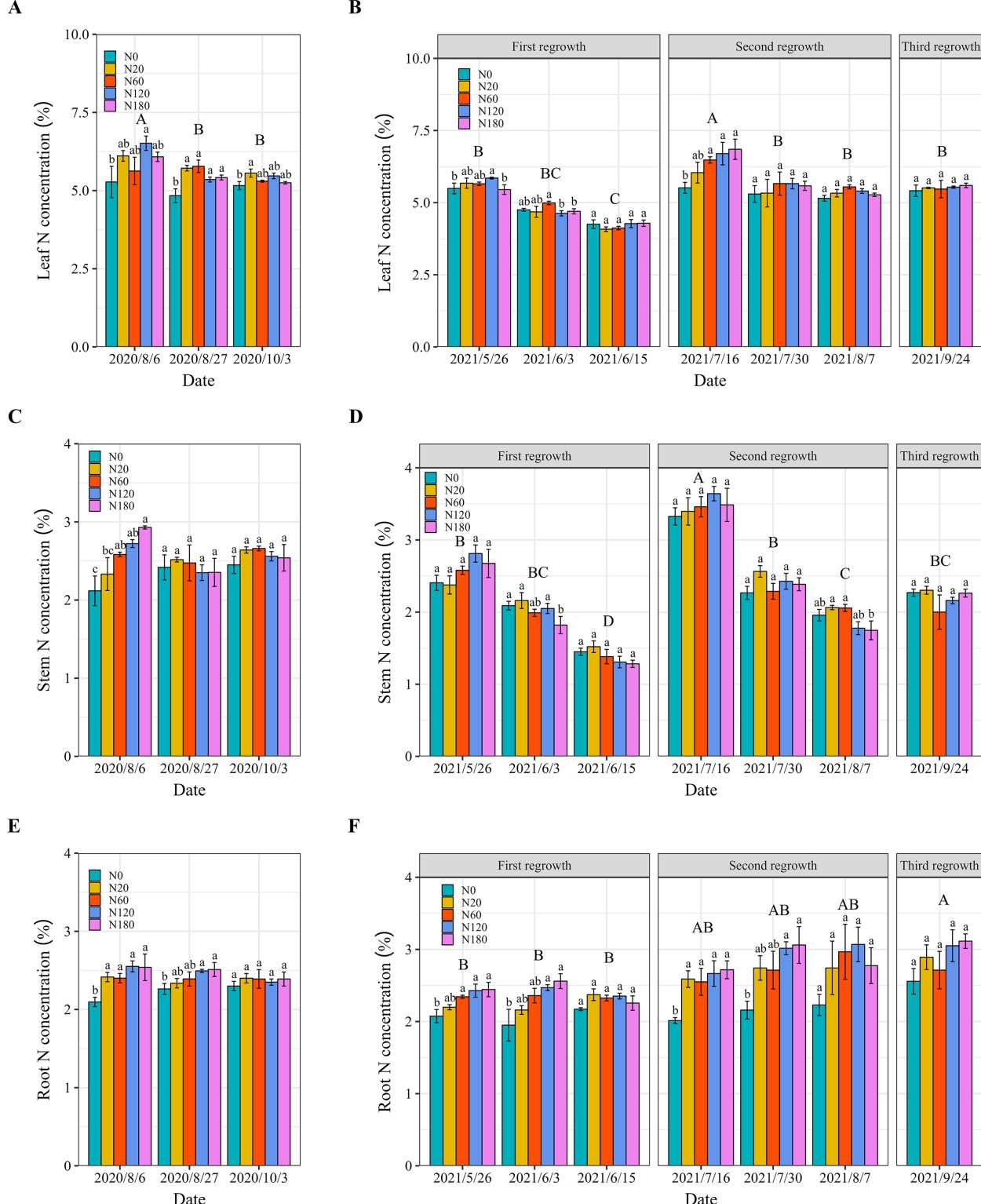

**Figure 5 Alfalfa leaf (A and B), stem (C and D) and root (E and F) N concentrations under different N application rates and on different dates in 2020 and 2021.** The data are presented as the means ± SEs ($n = 3$). The $P$ values for the main interactive effects of N fertilizer application rates (N) and date are reported in Tables S1 and S2. Different capital letters indicate significant differences among the dates. The vertical bars with different lowercase letters associated with each date indicate significant differences among the treatment means. The treatments and dates are the same as those described in Fig. 2.

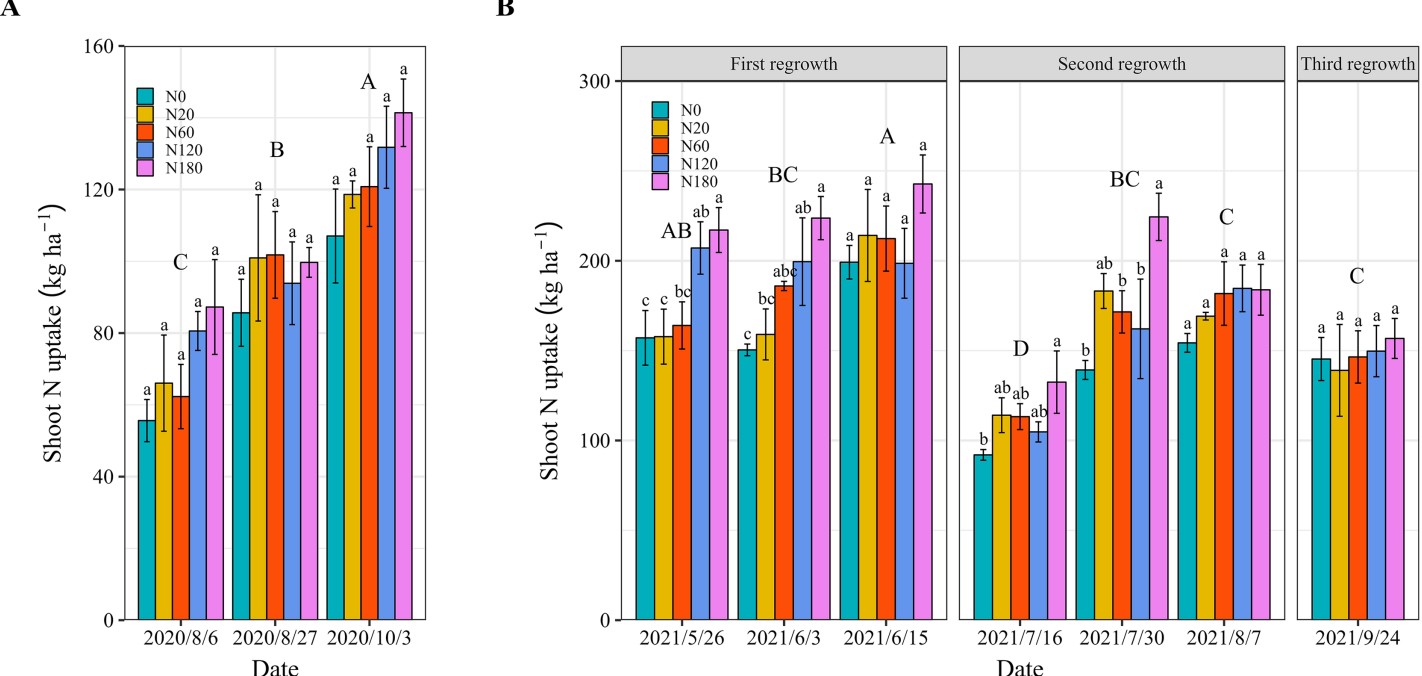

**Figure 6 Shoot N uptake of alfalfa under different N application rates and on different dates in 2020 (A) and 2021 (B).** The *P* values for the main interactive effects of N fertilizer application rates (N) and date are reported in Tables S1 and S2. Different capital letters indicate significant differences among the dates. The vertical bars with different lowercase letters associated with each date indicate significant differences among the treatment means. The treatments and dates are the same as those described in Fig. 2.

significant difference in the stem N concentration among the N fertilizer application rates on the other dates.

The root N concentration of 2020 did not change significantly among sampling dates. The N fertilizer application increased the root N concentration but not the leaf or stem N concentration (Fig. 5). The root N concentration increased by 10.2%–21.7% and 17.5%–38.1% compared to low N application rates in treatments with ≥120 kg N ha$^{-1}$ in 2020 and 2021, respectively. However, N fertilizer application did not lead to a further significant increase in the root N concentration at cutting (Figs. 5E and 5F).

There was no significant difference in alfalfa N uptake among the N fertilizer application rates in 2020 (Fig. 6A). In 2021, the shoot N uptake of first cut significantly higher than that of second and third cut. shoot N uptake increased by 19.0%–49.7% when N fertilizer application was greater than 120 kg N ha$^{-1}$ compared to that in N0 at the vegetative and bud stages (May 26, June 3, July 16 and July 30). However, N fertilizer application did not lead to further significant improvement in shoot N uptake at cutting (Fig. 6B).

## Alfalfa quality

Shoot CP concentration influenced by the main interactive effects of N fertilizer application and sampling date. The N fertilizer application improved the shoot CP concentration. In 2020, compared with that in N0, the shoot CP concentration in N fertilizer-treated plants significantly increased by 22.7% to 26.7% above 120 kg ha$^{-1}$ on August 6. With respect to alfalfa growth, low N application in N20 and N60 resulted in a

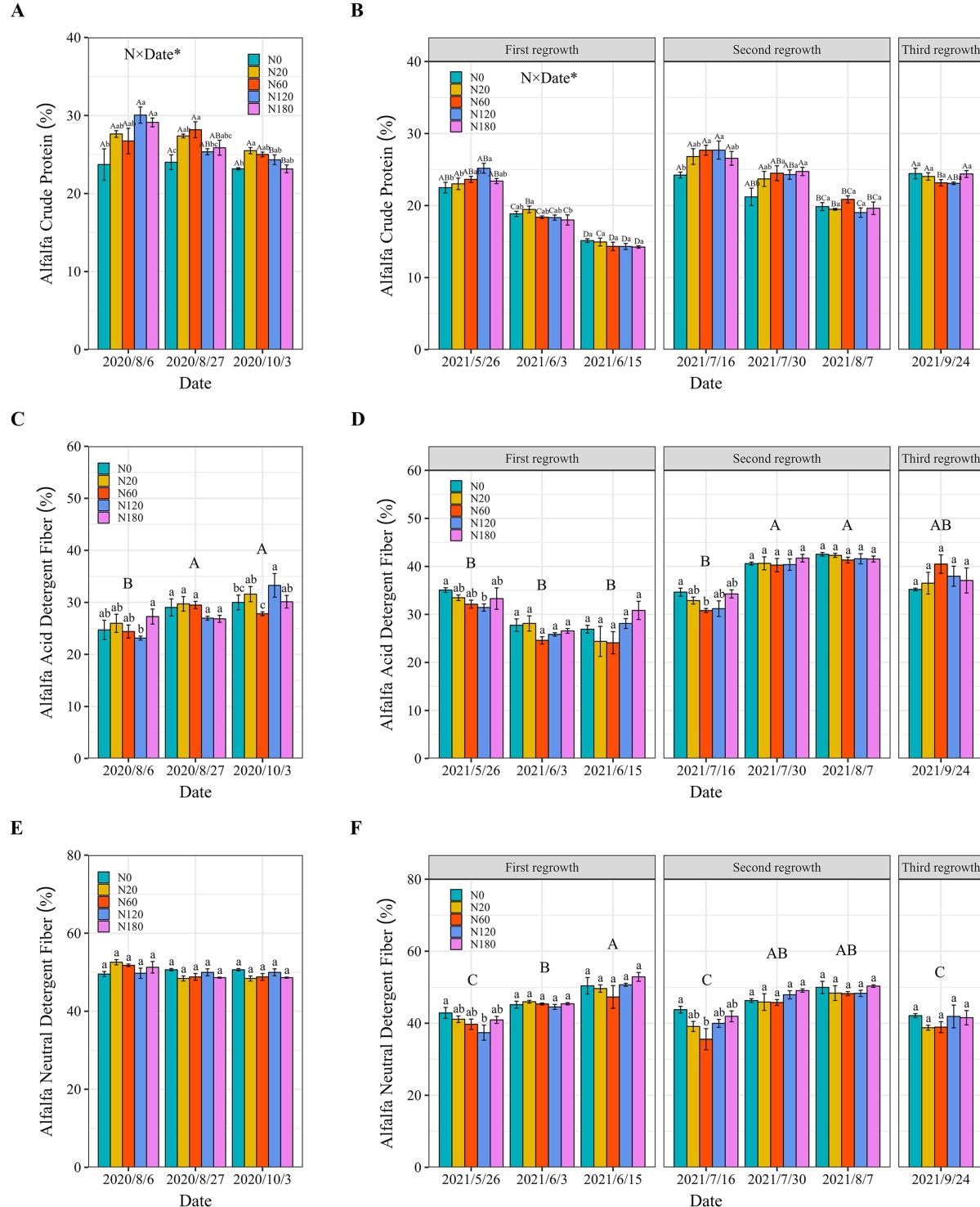

**Figure 7 Alfalfa crude protein (A and B), acid detergent fiber (C and D) and neutral detergent fiber (E and F) concentrations under different N application rates and on different dates in 2020 and 2021.** The data are presented as the means ± SEs (*n* = 3). The *P* values for the main interactive effects of N fertilizer application rates (N) and date are reported in Tables S1 and S2. Different capital letters indicate significant differences among the dates (Different capital letters for each bar indicate significant differences between different dates for the same treatment when the interaction is significant). The vertical bars with different lowercase letters associated with each date indicate significant differences among the treatment means. The treatments and dates are the same as those described in Fig. 2.

significant improvement of 10.0%–17.3% in the CP concentration compared to that in N0 on August 27 and October 3 (Fig. 7A). In 2021, compared with that in N0, the CP concentration in N120 significantly increased by 11.9% on May 26. However, this positive effect on the CP concentration was observed only under N20 compared to N180 on June 3. In addition, N fertilizer application led to a significant increase of 10.6%–15.5% in shoot CP concentration when the rate was above 60 kg ha$^{-1}$ compared to that in N0 on the second regrowth. However, there was no significant difference in the alfalfa CP concentration among the different N fertilizer application rates for any of the cuttings (June 15, July 30, and September 24) (Fig. 7B).

The N fertilizer application and sampling date consistently had a significant effect on shoot ADF and NDF concentration. In 2020, compared with that in N180, the shoot ADF concentration in N60 and N120 significantly decreased by approximately 10% on August 6, and October 3. However, no significant difference in NDF concentration was detected among the N fertilizer rates (Figs. 7C and 7E). In 2021, the shoot ADF concentration decreased by approximately 11% in N60 and N120 compared to that in N0 on May 26 and July 16. N application at rates of 60 kg N ha$^{-1}$ and 120 N kg ha$^{-1}$ also resulted in a significant decrease of 12.7%–18.7% in NDF concentration compared to that in N0. However, N fertilizer application did not further decrease the alfalfa ADF or NDF concentrations on other dates (Figs. 7D and 7F).

## Alfalfa nodule number, %Ndfa and amount of fixed N

In 2020, compared with other N fertilizer application rates, N applications of 20 and 60 kg N ha$^{-1}$ significantly increased the nodule number (Table 2). Similarly, the %Ndfa significantly decreased from 78.0% to 20.2% in N180 compared to that in N0. The corresponding amount of N fixed decreased N fixation by 65.6% regardless of the different N application rate. However, compared with the N application rates in N0 and N180, the application of 60 kg N ha$^{-1}$ significantly increased the nodule number and N fixation 2–3-fold and almost 1-fold, respectively, at the second cutting in 2021. In addition, the highest %Ndfa values of 53.8% and 68.6% were observed in N60 after both the second and third cuttings, respectively. However, no significant increase in alfalfa nodule number, %Ndfa, or N fixation was observed among the N fertilizer application rates on the other dates.

## Soil $NO_3^-$-N and $NH_4^+$-N concentrations

The main interactive effects of nitrogen fertilizer application and sampling date impacted soil $NO_3^-$-N and $NH_4^+$-N. The N fertilization significantly increased the concentrations of soil $NO_3^-$-N and $NH_4^+$-N. The soil $NO_3^-$-N concentration was greater than the $NH_4^+$-N concentration in both 2020 and 2021 (Fig. 8). In 2020, compared with that in N0, the soil $NH_4^+$-N concentration in N120 significantly increased by 24.1%–33.3% (Fig. 8A). In contrast, the increase in soil $NO_3^-$-N was greater than that in soil $NH_4^+$-N in the treatments with N fertilizer. Compared with that in N0, the soil $NO_3^-$-N concentration significantly increased 1–3-fold when the fertilization rate was greater than 120 kg N ha$^{-1}$ (Fig. 8B). In 2021, compared with that in N0, the soil $NH_4^+$-N concentration was significantly lower on June 15, July 16 and August 7 in response to N fertilizer application that exceeded

**Table 2  Number of nodules, % Ndfa and amount of N fixed at different N fertilizer application rates.**

| Date | Treatment | No. of nodule | % Ndfa (%) | Amount of N fixed (kg ha$^{-1}$) |
|---|---|---|---|---|
| 2020/10/3 | N0 | 5.2 ± 0.7 b | 78.0 ± 16.8 a | 81.1 ± 2.3 a |
| | N20 | 10.8 ± 1.3 a | 60.4 ± 16.9 a | 72.0 ± 12.7 a |
| | N60 | 10.2 ± 1.4 a | 55.0 ± 12.7 a | 65.3 ± 6.8 a |
| | N120 | 4.3 ± 1.1 b | 51.5 ± 26.4 ab | 55.9 ± 12.5 ab |
| | N180 | 3.8 ± 1.2 b | 20.2 ± 8.9 b | 27.7 ± 5.6 b |
| 2021/6/15 | N0 | 4.9 ± 2.4 a | 34.9 ± 15.2 a | 67.8 ± 27.1 a |
| | N20 | 2.1 ± 0.6 a | 36.8 ± 11.3 a | 73.2 ± 14.0 a |
| | N60 | 4.9 ± 0.5 a | 47.5 ± 7.6 a | 98.1 ± 6.7 a |
| | N120 | 2.9 ± 1.5 a | 36.6 ± 4.4 a | 71.4 ± 4.5 a |
| | N180 | 1.8 ± 0.7 a | 32.5 ± 5.3 a | 79.3 ± 8.3 a |
| 2021/8/7 | N0 | 9.0 ± 2.0 bc | 31.2 ± 9.5 ab | 46.3 ± 14.5 b |
| | N20 | 14.7 ± 0.6 ab | 35.7 ± 6.7 ab | 55.8 ± 8.7 b |
| | N60 | 16.6 ± 1.5 a | 53.8 ± 9.7 a | 95.8 ± 12.9 a |
| | N120 | 9.8 ± 2.8 bc | 32.7 ± 7.0 ab | 50.6 ± 2.9 b |
| | N180 | 5.3 ± 5.3 c | 25.4 ± 4.1 b | 46.1 ± 5.3 b |
| 2021/9/24 | N0 | 14.2 ± 2.1 a | 46.5 ± 2.9 b | 67.1 ± 4.2 a |
| | N20 | 20.4 ± 6.3 a | 64.1 ± 5.3 ab | 89.5 ± 17.9 a |
| | N60 | 17.6 ± 8.8 a | 68.6 ± 6.9 a | 101.1 ± 16.0 a |
| | N120 | 14.6 ± 9.3 a | 64.5 ± 6.4 ab | 97.8 ± 17.8 a |
| | N180 | 12.0 ± 6.5 a | 57.9 ± 6.8 ab | 92.3 ± 16.6 a |

**Note:**
Values are the means ($n = 3$) ± SEs. The different letters indicate significant differences among the N fertilizer application rates on the same date.

120 kg ha$^{-1}$ (Fig. 8C). In contrast, the soil $NO_3^-$-N concentration significantly increased by 1 to 3 times when the N fertilizer application rate was greater than 120 kg N ha$^{-1}$ compared to that in N0 (Fig. 8D). However, N fertilizer application did not lead to further increases in the soil $NO_3^-$-N and $NH_4^+$-N concentrations at the third regrowth, and this was particularly evident for the soil $NO_3^-$-N concentration in the 20–40 cm soil layer.

## DISCUSSION

Previous studies have shown that N fertilizer application cannot increase alfalfa yield and can even reduce yield because N fixation supplies sufficient N for alfalfa (*Malhi et al., 1992*; *He, Xie & Li, 2018*; *Lindström & Mousavi, 2020*). In this study, the application of N fertilizer did not improve alfalfa growth, as indicated by the height, stem diameter and branch number, except in N180 (Figs. 2 and 3). When nodules are functioning effectively, additional N from fertilizers may have a limited impact on growth parameters like height, stem diameter, and branch number, which is consistent with the findings of *Zhang et al. (2017)*. A high N application rate improved alfalfa growth in salt-affected soil. It was possibly due to that N fertilizer application causes high chlorophyll content and photosynthetic capacity of leaves in salt-affected soil (*Fan et al., 2016*; *Kamran et al., 2022*). However, high N rates can increase input costs for farmers, which may not always be

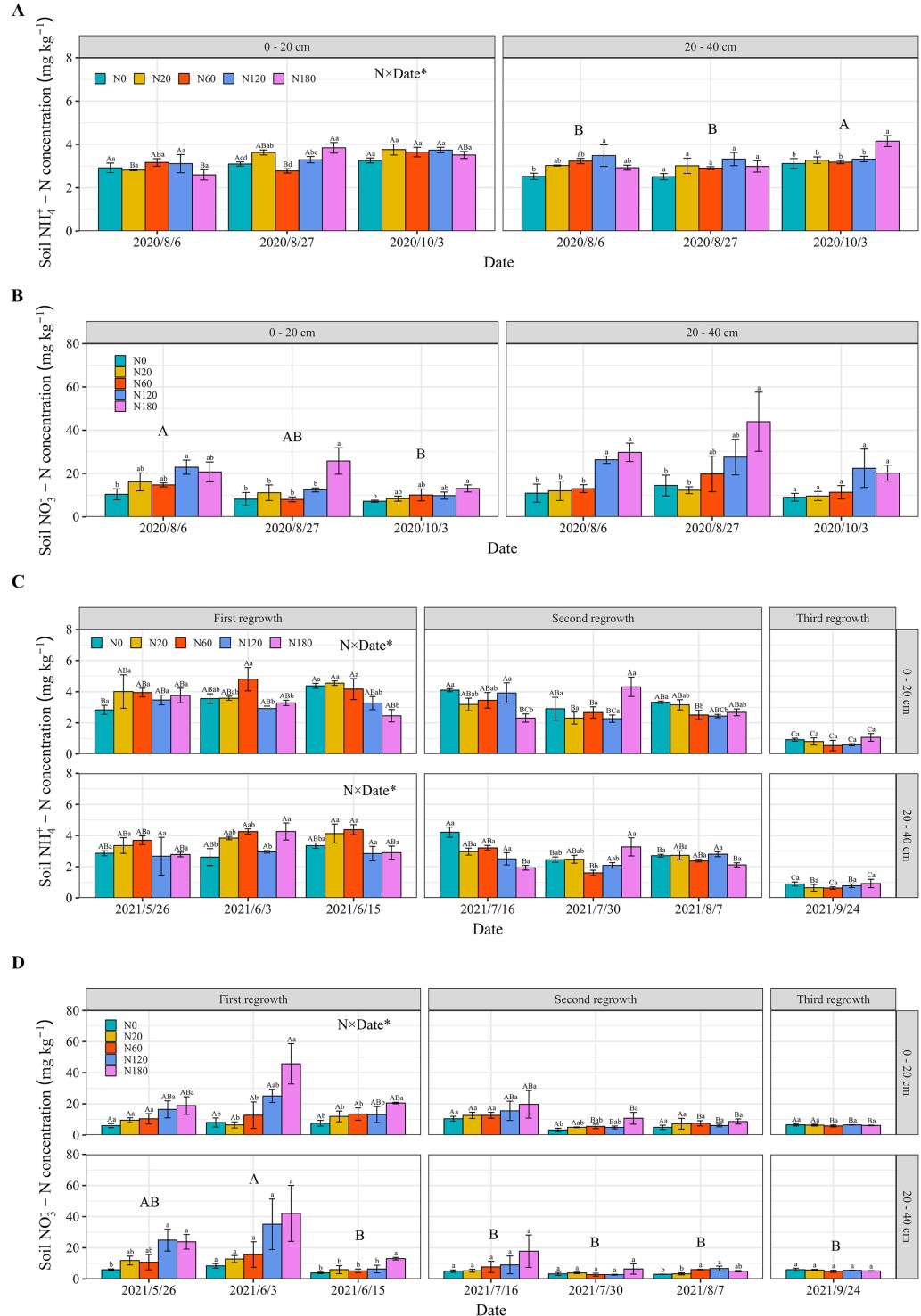

**Figure 8 Soil NH$_4^+$-N concentration (A and C) and NO$_3^-$-N concentration (B and D) in the different soil layers under the different N application rates and on different dates in 2020 and 2021.** The data are presented as the means ± SEs (*n* = 3). The *P* values for the main interactive effects of N fertilizer application rates (N) and date are reported in Tables S1 and S2. Different capital letters indicate significant differences among the dates (Different capital letters for each bar indicate significant differences between different dates for the same treatment when the interaction is significant). The vertical bars with different lowercase letters associated with each date indicate significant differences among the treatment means. The treatments and dates are the same as those described in Fig. 2.

justified by the yield increase, especially if the economic value of alfalfa or other environmental conditions change. The results also showed that soil N significantly increased when N fertilizer was applied at a rate that exceeded 120 kg N ha$^{-1}$ (Fig. 8). A balanced approach that considers both economic viability and environmental sustainability is critical.

The N application has been found to improve the N concentration of alfalfa by increasing the leaf number (Slamet et al., 2012). Our findings confirmed that N application rates of 60 or 120 kg N ha$^{-1}$ significantly increased the leaf N concentration compared with that in N0 (Figs. 5A and 5B). In contrast, N fertilizer application did not change the stem N concentration. This is because leaves are the primary site for nutrient uptake and photosynthesis in plants, and alfalfa growth and development typically rely more on the development of leaves (Ciampitti & Vyn, 2011). As discussed above, the increase in leaf growth also contributes to this result. However, N fertilizer application rates that exceeded 120 kg ha$^{-1}$ did not result in further increases in the N concentration. This limitation is due to the restricted N uptake capacity, leading to lower N fertilizer utilization efficiency (Wang et al., 2011; Xu et al., 2012; Ye et al., 2013). In both 2020 and 2021, the root N concentration increased more than the leaf and stem N concentrations in the N fertilizer application treatments, indicating that N allocation was prioritized (Fig. 5). These findings are also consistent with those of other studies (Ramasamy, ten Berge & Purushothaman, 1997; Chen et al., 2020). In addition, our findings align with previous studies showing that N fertilizer significantly increases shoot N uptake in both legumes and nonleguminous plants (Wortmann, McIntyre & Kaizzi, 2000; Abbasi et al., 2011). In this study, although it was not possible to determine the N uptake of the entire root system due to incomplete sampling of alfalfa roots in the field, the results still indicate that alfalfa removed a large amount of N from the soil, even in N0. In addition to N fixation by nodules, mineralization of soil organic matter is an important source of N uptake in alfalfa (Kuiters, 1990; Weintraub & Schimel, 2003; Cheng & Kuzyakov, 2015; Frouz, 2018). Future studies should incorporate more comprehensive root sampling and mineralization N of different soil layers to better understand how N is distributed between different plant tissues and soil layers.

Several studies have shown that direct N fertilization does not increase the forage CP concentration (Oliveira et al., 2004). In contrast, the results showed that N application rates of 60 and 120 kg ha$^{-1}$ significantly increased the alfalfa CP concentration in 2020 and in the vegetative stage and bud stage of 2021 (Fig. 7). This suggested that N fertilizer application alleviated the negative impact on shoot CP concentration caused by salt stress, as documented by Wan et al. (2023). This improvement became weak at the early flowering stage, which may be due to the retranslocation of N from vegetative tissues to reproductive tissues (Moot et al., 2015). In saline–alkali soil, the uptake and translocation of nutrients may be inhibited at the reproductive tissue stage, which can limit the beneficial effects of N fertilization on forage quality (Parida & Das, 2005). A larger root area and longer root length were found in the second year. Further research is needed to explore and fully explain the transfer of N between shoots and roots and the N supply dynamics in deeper soil layers. The decrease in cell wall concentrations caused by N fertilizer

application may have been responsible for the decreases in ADF and NDF (*Parsons, Cherney & Peterson, 2009*). Similar to the shoot CP concentration, ADF and NDF did not decrease further at the early flowering stage.

Previous studies have shown a negative exponential connection between N fertilizer rates and N fixation in legumes, driven by plant preferences for acquiring N from the soil (*Sanginga, 2003*; *Salvagiotti et al., 2008*). This study confirmed that excessive N inhibited the increase in nodule number and decreased total N fixation. Notably, fertilizer applications of 20 and 60 kg N ha$^{-1}$ increased the nodule number, %Ndfa, and amount of N fixed. This was due to salinity may decreased nodule formation, and further lead to the negative effect on the rhizobia colonization in the root (*Del Pilar Cordovilla, Ligero & Lluch, 1999*; *Arora, 2015*). The N supply supports nodule initiation and enhances nitrogen fixation by rhizobia by boosting nitrogenase and nitrate reductase activity in legumes grown in low-fertility, salt-affected soils (*Abdel Wahab & Abd Alla, 1995*; *Undersander, 2011*; *Elgharably & Benes, 2021*; *Liu et al., 2022*). However, there was no difference in nodule number among the N application rates in the first and third cuttings of 2021. The higher %Ndfa in N60 indicated that the activity and size of the nodules may be limited in N0 by carbohydrate deprivation and its impact on the size of carbohydrate pools within the nodules (*Parsons et al., 1993*; *Jeudy et al., 2010*). Overall, the optimal N fertilizer application rate was 60 kg N ha$^{-1}$ for alfalfa production in salt-affected soils, at least for N fixation. This finding is crucial for China alfalfa production in saline–alkali soils, as it provides a guidance to optimizie N fertilizer application. But it should be note that these results were obtained in a saline–alkali soil. Therefore, the findings may not be fully applicable to other soil types.

This study showed that N fertilizer application significantly increased the soil $NO_3^-$-N and $NH_4^+$-N concentrations at depths of 0–20 cm and 20–40 cm in N120 and N180 (Fig. 8). These findings are consistent with those of previous studies, which have shown that the application of N fertilizer is linearly associated with soil $NO_3^-$-N and $NH_4^+$-N concentrations (*Tomer & Burkart, 2003*; *Cambouris et al., 2017*). In addition, the soil $NH_4^+$-N concentration significantly decreased when N fertilizer application exceeded 120 kg ha$^{-1}$ on some dates in 2021. This was due to improved nitrification resulting from an increase in the soil $NH_4^+$-N concentration, indicating that more soil $NH_4^+$-N was converted to $NO_3^-$-N when the N fertilizer application was greater than 120 kg N ha$^{-1}$ (*Powlson, 1993*; *Schmitt, Sheaffer & Randall, 1994*). The efficiency of N fertilizers declines during later periods because of N fertilizer loss over time (*Soon & Malhi, 2005*; *Ju et al., 2009*). These findings aligned with this study (Fig. 8). Therefore, it is crucial to explore the long-term implications of N fertilizer application for soil health and groundwater quality. Excessive N application can lead to increased N losses, such as ammonium emission and nitrate leaching (*Xu et al., 2012*; *Ye et al., 2013*). These losses not only harm soil health by disrupting nutrient balances but also contaminate groundwater and contribute to aquatic eutrophication (*Cameron, Di & Moir, 2013*). To reduce environmental risks, the N fertilizer application rate for farmers should not exceed 120 kg ha$^{-1}$. Alfalfa is a perennial forage grass, and preliminary findings were obtained from this 2-year field research. Our study also confirmed that the interaction of date and N fertilizer application rate

significantly affects soil N. Thus, environmental variability, including yearly differences in rainfall and soil conditions, could influence alfalfa biomass and yield components differently across time (*Feng et al., 2022*). Thus, there is a clear need for long-term exploration in the field of alfalfa yield, quality, N nutrition and soil N research.

## CONCLUSIONS

This study revealed the essential role of N fertilizer application at low rates in alfalfa production, as this practice not only increases alfalfa yield but also improves N fixation in saline–alkali soil. However, it did not result in further improvement in alfalfa quality at the early flowering stage. Furthermore, we have also shown that excessive N fertilizer application results in an increase in soil $NO_3^--N$ and $NH_4^+-N$, which may increase the risk of N leaching or ammonia volatilization from soil. These findings offer valuable insights for N management practices for alfalfa in saline–alkali soils. Farmers should recognize the importance of N fertilizer while balance yield benefits with environmental sustainability. And further research is needed to explore how N nutrition in alfalfa is influenced by different soil types and environmental conditions.

### Funding

This work was funded by the National Natural Science Program of China (2023YFD1900404). The funders had no role in study design, data collection and analysis, decision to publish, or preparation of the manuscript.

### Grant Disclosures

The following grant information was disclosed by the authors:
National Natural Science Program of China: 2023YFD1900404.

### Competing Interests

The authors declare that they have no competing interests.

### Author Contributions

- Weifan Wan conceived and designed the experiments, performed the experiments, analyzed the data, prepared figures and/or tables, authored or reviewed drafts of the article, and approved the final draft.
- Qian Liu performed the experiments, prepared figures and/or tables, and approved the final draft.
- Ke Li performed the experiments, prepared figures and/or tables, and approved the final draft.
- Kun Zhao performed the experiments, prepared figures and/or tables, and approved the final draft.
- Fei Qi performed the experiments, prepared figures and/or tables, and approved the final draft.

- Yuanshuo Li performed the experiments, prepared figures and/or tables, and approved the final draft.
- Zhi Sun performed the experiments, prepared figures and/or tables, and approved the final draft.
- Haigang Li conceived and designed the experiments, authored or reviewed drafts of the article, and approved the final draft.

## Data Availability

The raw data and code for figures are available in the Supplemental Files.

## Supplemental Information

Supplemental information for this article can be found online at http://dx.doi.org/10.7717/peerj.18796#supplemental-information.

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
