# Peer review of "Nitrogen fertilizer application for improving the biomass, quality, and nitrogen fixation of alfalfa (Medicago sativa L.) at different growth stages in a saline‒alkali soil"

_PeerJ, doi:10.7717/peerj.18796_

## Round 0.1 · original submission · Major Revisions

Dear Authors

The manuscript cannot be accepted for publication in its current form. It needs a major revision before publication. The authors are invited to revise the paper considering all the suggestions made by the reviewers. Please note that the requested changes are required for publication.

With Thanks

Reviewer 1 ·

Basic reporting

Thank you for considering me for reviewing the manuscript “Nitrogen fertilizer application for improving the biomass, quality, and nitrogen fixation of alfalfa (Medicago sativa L.) at different growth stages in a saline-alkali soil”. The research addresses an important agricultural issue, particularly in regions with poor soil quality.

Suggestion:
Conduct a thorough English language edit of the manuscript to correct grammatical errors and improve clarity. Summarize redundant sentences and focus on highlighting the important aspects of the study.

The abstract is generally well-written providing a clear summary of the study aims. However, it could be improved by including specific numerical results for better impact and clarity. Clarify relationship between nitrogen levels and plant productivity and quality in the abstract. Highlight the broader implications of the study results for agricultural practices in saline-alkali soils.

Include relevant keywords that are not presented in the title and abstract to improve searchability.

The introduction should be improved, highlight solid background on the challenges associated with saline-alkali soils, and the rationale for studying nitrogen fertilizer application. The knowledge gap should be emphasized, and the novelty of the study. State the novelty of the research, i.e., what differentiates this study from previous research on nitrogen fertilization in alfalfa. Add references to recent studies that highlight the current trends or contrasting results in the field. The hypothesis and objectives need to be clearly defined.

The methodology has an appropriate description of experimental setup, nitrogen treatments, data collection, and statistical analysis. However, the specific nitrogen application rates require further justification. This section could benefit from visual aids like diagrams or schematics of the experimental design for better visualization and understanding.
Line 92 needs more details on the experimental soil. More information on soil chemical and physical properties should be clarified. Soil particle distribution percentages of clay, silt, and sandy, electrical conductivity, soluble cations, soluble anions, and available nutrients should be clarified.

The Results section requires significant revision and enhancement. It should begin with a thorough analysis of variance (ANOVA) to establish a foundational understanding of data variability and statistical significance. This will provide readers with a clear context for the subsequent findings.
The interpretation of the results needs to be more precise, particularly regarding their practical implications. Emphasize the real-world significance of the findings, such as the extent to which yield improvements can impact agricultural practices and productivity. Highlight how these results can be applied in practical scenarios, especially for farmers dealing with saline-alkali soils.
The resolution of Figures 1-7 is currently inadequate, leading to unclear and pixelated images. For electronic publications, these figures must be clear and legible at 100% zoom.
Estimating and presenting nitrogen use efficiencies is recommended. Including this data will provide a more comprehensive understanding of how nitrogen application influences alfalfa productivity, contributing valuable insights to the study practical applications.

The discussion section would benefit from a more critical analysis of the study limitations. The manuscript briefly explores these limitations but does not thoroughly explore how they might influence the findings or the broader implications of the research. A deeper examination of the study constraints, particularly concerning environmental variability and the specific conditions of the experimental site, is essential. Furthermore, future research directions need to be more comprehensive. In particular, consider how these findings could be applied to different regions or soil types and suggest specific areas where further investigation is needed to validate and extend the study conclusions. Use updated citations; there are references from 1992 that are not recommended.

The conclusion could be improved by reinforcing the study broader contributions to agricultural practices, particularly in terms of how these findings can influence nitrogen management practices in saline-alkali soils worldwide. Suggest potential policy implications or practical guidelines for farmers based on the study findings.

For uniformity and accuracy, follow the journal style guide or citation requirements. Revise the journal abbreviations in the references for consistency.
Lines 384 and 521: Scientific names should be in italics throughout the manuscript.

Experimental design

The experimental design is appropriate

Validity of the findings

The findings presented in the manuscript are valid, supported by robust data and sound statistical analysis.

Reviewer 2 ·

Basic reporting

No comment

Experimental design

The study demonstrates an acceptable experimental design.

Validity of the findings

No comment

Additional comments

The study provides compelling evidence that N fertilizer application can significantly enhance alfalfa yield and quality in saline-alkali soils. By addressing a critical knowledge gap, the researchers offer valuable insights for agricultural practices in these challenging environments. The findings suggest that N fertilizer application is a promising strategy for improving alfalfa production, providing valuable guidance for farmers and agricultural policymakers seeking to optimize crop yields and nutritional value in marginal lands.
-Comments and Suggestions for Authors
- Abstract
-The abstract provides a clear and concise overview of the research.
- Consider adding a brief discussion of the implications of the findings. For example, what are the practical implications for farmers and policymakers?
- Introduction
- Consider adding a brief overview of N fixation in alfalfa. This would help readers who may not be familiar with the process.
- Expand on the potential negative impacts of excessive N fertilizer application. This could include references to environmental concerns and potential harm to livestock.
- Materials & Methods
- The materials and methods section is well-structured and provides a clear description of the experimental procedures. It includes essential details about the experimental site, design, sampling, and analytical methods.
-The use of SAS for data analysis is appropriate. The choice of the least significant difference (LSD) test is a common and valid method for comparing means.
- Explain the rationale for choosing the specific N fertilizer application rates. This would help to justify the range of treatments used.
- Results
- The results section provides a comprehensive and well-structured analysis of the impact of N fertilizer application on alfalfa. The authors effectively present their findings, supported by clear and informative figures.
- Did the authors take in account the weather conditions during the experiment, including precipitation, temperature, and humidity? This would allow readers to assess the potential impact of weather variability on the results.
- Did the authors explored the potential interactions between N fertilizer application and other environmental factors such as salinity and alkalinity, water availability?
- Discussion
- The discussion section provides a valuable contribution to the understanding of N fertilizer management in alfalfa production on saline-alkali soils.
- Remove the subtitles in discussion section.
Consider discussing the potential interactions between N fertilizer application and other environmental factors. This could include exploring the effects of salinity, alkalinity, and water availability on the response of alfalfa to N fertilizer.
-The discussion could provide more specific recommendations for farmers regarding N fertilizer application in saline-alkali soils.
- Conclusion
-Consider adding a brief statement about the potential long-term implications of N fertilizer application. This could include discussing the potential impacts on soil health and sustainability.

Reviewer 3 ·

Basic reporting

The manuscript presents a study investigating the effects of nitrogen (N) fertilizer application on alfalfa yield, quality, and N fixation in saline-alkali soils. The research is well-structured and addresses an important topic in sustainable agriculture. The experimental design is appropriate, with clear descriptions of the treatments, sampling procedures, and analytical methods used. However, minor improvements could be made in explaining the rationale behind specific methodological choices, such as the selection of N application rates. Result section is appropriate with key values of parameters under study. The Discussion section highlights key findings, such as the positive impact of low-rate N fertilizer application on alfalfa yield and N fixation, while also addressing the limitations in improving forage quality at the early flowering stage. The discussion of environmental risks associated with excessive N application is a critical aspect, though the section could benefit from a more comprehensive exploration of the broader implications and recommendations for future research. The Conclusion section could be strengthened by providing more specific recommendations for future research and discussing the practical implications of the findings for agricultural practices. The manuscript is well-written and makes a valuable contribution to the field. With minor revisions, particularly in expanding the discussion and conclusion sections, it has the potential to provide significant insights into sustainable alfalfa production in saline-alkali soils.

Specific Comments
Introduction: A more concentrated review of literature directly related to the study’s primary focus would strengthen the introduction. Additionally, the introduction must delve deeper into the mechanisms by which N fertilization affects alfalfa growth and quality, especially under suboptimal conditions. Please add the hypotheses being tested in this section. Incorporate more recent advances in alfalfa cultivation or related technologies, such as precision agriculture, that could complement traditional fertilization strategies for broader context of modern agricultural practices. Please address why the specific N application rates (0, 20, 60, 120, 180 kg N ha-1) were chosen. Are these rates based on previous studies, local farming practices, or specific hypotheses about N response in alfalfa? While the introduction effectively addresses the national context (China’s alfalfa production), it could benefit from a brief discussion on the broader implications of the study’s findings. For example, how might the results inform global practices in forage production, particularly in regions with similar environmental challenges?
Line No. 48 “(Raun et al., 1999; Xie et al., 2015)” please cite recent literature.
Materials and Methods
Please provide complete environmental data of the study site for two years. Add soil moisture content, rainfall along with temperature to indicate their effects on alfalfa growth and nitrogen dynamics and to comprehensively understand the experimental conditions. While the soil is classified as "light soda-alkalized aqueous soil", it would be beneficial to provide electrical conductivity (EC) and cation exchange capacity (CEC) in Table 1 and results to understand the baseline soil fertility and the potential impact of N fertilization. The plot size of 60 m² is mentioned, but it would be helpful to discuss whether this size is sufficient to capture the variability within and between treatments. Additionally, any potential edge effects or interactions between adjacent plots should be addressed, as they could affect the validity of the results. In Line No. 11 authors mentioned that “No herbicides and insecticides measures were taken in 2021.” Explain why?
Line No. 123 “sample was separated into stems and leaves, which were weighed and oven-dried at 105 °C for 0.5 h” What do you mean by 0.5 hours. Please check is it correct. Mention in minutes.
The Van Soest method is mentioned for determining neutral detergent fiber (NDF) and acid detergent fiber (ADF) concentrations, but a brief description of the procedure or reference to a detailed protocol would be useful for readers unfamiliar with this technique. Although the irrigation schedule is described, more details on the total amount of water applied and the method used to ensure uniform water distribution across plots would be beneficial. This is especially important in a semiarid region where water availability can significantly influence experimental outcomes.
Figures were produced in R studio using which tools? Mention. Also Provide R coding used to draw figures.
Discussion: Some sentences are complex and could be simplified for better clarity. For example, the sentence (in Line No. 277-278) "This limitation can be attributed to the constrained N uptake capacity, which results in relatively low N fertilizer utilization efficiency" could be rephrased to enhance readability. While the discussion provides an overview of the impact of N fertilizer on yield components such as height, stem diameter, and branch number, it could delve deeper into why these specific components did not respond significantly to most N application rates. For instance, discussing potential physiological or environmental constraints on these components could add depth to the analysis. The discussion lacks consideration of the economic and practical implications of the findings. For example, while the manuscript suggests that a high N application rate is necessary for alfalfa growth in poor soils, it does not discuss the cost-effectiveness or sustainability of such an approach. Including this analysis would make the discussion more relevant to farmers and policymakers. The manuscript states that (Line No. 262-263) "a high N application rate was necessary when alfalfa was grown in poor soils, such as salt-affected soil," based on the study's findings. However, this statement might be too broad, as it does not fully account for variations in soil type, climate, and management practices. It would be more appropriate to suggest that further research is needed to generalize these findings to other conditions. Line No. 263 avoid to use terms like “our results”. Line No. 265 “N fertilizer facilitates the improvement of leaf chlorophyll content” avoid to start a sentence with symbols, abbreviation or number. The manuscript briefly mentions the potential for N losses, such as nitrate leaching, when N fertilizer application exceeds 120 kg N ha-1. However, this important environmental concern could be discussed in more detail. The discussion could explore the long-term implications of such losses for soil health and groundwater quality, making the findings more relevant to discussions on sustainable agriculture. Line No. 325 “(M. D. Tomer, 2003” correct intext citation style.
Conclusion: The conclusion could be significantly strengthened by expanding on the broader implications of the findings, providing specific recommendations for future research or agricultural practices, acknowledging the study's limitations, and offering a more cohesive and integrated summary.

The current assignment of letters to bars using the LSD test in R seems incorrect. Specifically, bars with very small differences, which do not even overlap in standard errors (SEs), are assigned the same letter as significantly larger bars. This misrepresentation could mislead readers into interpreting significant differences as non-significant, thereby questioning the validity of the statistical analysis. Figure 1-7: Multiple instances exist where the smallest bars, which should be statistically distinct from others due to non-overlapping SEs, are given the same letter as the largest bars. This inconsistency suggests that the LSD post-hoc test has not been appropriately applied or visualized in these figures. This issue is not just a minor technicality; it directly affects the interpretability and credibility of the study’s results. By assigning the same letters to bars that likely differ significantly, the figures may inaccurately communicate the presence of significant differences to the readers. As a result, the conclusions drawn from these figures could be flawed. It is crucial to re-run the LSD post-hoc tests with a focus on ensuring that the assignment of letters reflects actual statistical differences. This might involve checking the parameters used in R and ensuring that the comparison considers the correct alpha level. If the issue persists, consider using alternative software for post-hoc analysis and visualization, such as GraphPad Prism, SigmaPlot, or SAS, which might provide more accurate or user-friendly representations of LSD results. After reassigning the letters, visually inspect the figures to ensure that bars with overlapping SEs do not share letters with bars that have significantly different values. By addressing these concerns, the figures will more accurately reflect the statistical analysis, thereby enhancing the robustness and credibility of the study.
To ensure clarity and enhance the self-explanatory nature of the figures and tables, it is essential to include the full forms of all abbreviations used, specifically for CP (Crude Protein), ADF (Acid Detergent Fiber), and NDF (Neutral Detergent Fiber). Including these definitions directly in the captions will prevent any potential ambiguity for readers unfamiliar with these terms and improve the overall readability of the manuscript.

Experimental design

Experimental design of the study is appropriate.

Validity of the findings

The validity of the reported findings is currently compromised due to the inappropriate application of Least Significant Difference (LSD) lettering in the figures. The current LSD lettering scheme inaccurately represents significant differences, leading to instances where results that are statistically significant are incorrectly marked as non-significant. This misrepresentation undermines the reliability of the conclusions drawn from the data. It is imperative that the authors address this issue with urgency by re-analyzing the data and applying correct LSD lettering. This correction is crucial to ensure that the visual representation of the data accurately reflects the statistical significance of the results, thereby maintaining the integrity and credibility of the study.

---

## Round 0.2 · Major Revisions

Dear Authors
The manuscript cannot be accepted for publication in its current form. It needs a major revision before publication. The authors are invited to revise the paper considering all the suggestions made by the reviewers. Please note that the requested changes are required for publication.
With Thanks

Reviewer 1 ·

Basic reporting

The authors have made a thorough effort to address the comments raised in the previous review. I have carefully reviewed the revised manuscript, and it is evident that the authors have incorporated the feedback constructively. The revisions have improved the clarity and overall quality of the manuscript.

Experimental design

The experimental design is appropriate

Validity of the findings

The findings presented in the manuscript are valid, supported by robust data and sound statistical analysis.

Reviewer 2 ·

Basic reporting

no comment

Experimental design

no comment

Validity of the findings

no comment

Additional comments

The authors have made the changes I suggested in the last review. I recommend its publication in this journal

Reviewer 3 ·

Basic reporting

I am pleased to see authors have addressed most of the queries but there are still multiple typos in the article Like in Line No. 96-97 “Additionally, high nitrate levels in forage can nitrate toxicity of livestock (Kellems & David, 2002).” Line No.212-213 “Figures were generated using the ggplot, cowplot, rstatix, and car packages in RStudio.in RStudio software (version 4.2.1)” Line No. 336-337 “A high N application rate improved alfalfa growth in p salt-affected soil.” What do you mean by p here? Please carefully check and correct the typos in the article.
Thank you for addressing the concerns regarding the statistical analysis and rechecking the results using both R and SAS. However, despite the reanalysis, the assignment of letters in the figures remains problematic. The main concern is not just about SE overlap but about accurately reflecting statistical significance. The current letter assignment still does not align with the observed differences, especially in cases where visibly distinct bars are assigned the same letter. It is recommended to apply a two-way ANOVA to assess the significance across both nitrogen application rates and sampling dates. Treat the data in Figures 2–8 as a cohesive dataset, and ensure that the letter assignments reflect accurate statistical significance. Without proper correction, the current letter assignment may compromise the scientific integrity of the results by misrepresenting the actual statistical differences.

Experimental design

Experimental design is promising.

Validity of the findings

The validity of the results is uncertain due to inadequacies in the statistical analysis. Without the application of proper statistical methods, the findings cannot be considered robust and are unsuitable for presentation on a scientific platform. It is strongly recommended that the authors reanalyze the data presented in Figures 2-8, ensuring correct statistical procedures are followed. The results section should then be revised and reinterpreted based on this updated analysis.

---

## Round 0.3 · Minor Revisions

Dear Authors
The manuscript still needs a minor revision before publication. The authors are invited to revise the paper considering all the suggestions made by the reviewer. Please note that the requested changes are required for publication.
With Thanks

Reviewer 3 ·

Basic reporting

Thank you for revisiting the statistical analysis and addressing previous concerns by reanalyzing the data using both R and SAS. However, the improvements in figure lettering still require attention. The assignment of letters in Figures 2–8 remains problematic. In some sub-figures (or the parts of sub-figures) lettering is not assigned to nitrogen fertilization. Please assign letters to all sub-figures completely. Without proper corrections, the current presentation risks misrepresenting the data and may compromise the validity of the findings. I look forward to reviewing the revised figures and further supporting the manuscript's improvements.

Experimental design

Experimental design seems promising.

Validity of the findings

The validity of the findings can only be assessed after satisfactory improvements are made in the figure presentation, ensuring that the statistical analyses and their visual representation are accurately aligned.

---

## Round 0.4 · accepted · Accept

Dear Authors,

I am pleased to inform you that the manuscript has improved after the last revision and can be accepted for publication.

Congratulations on accepting your manuscript, and thank you for your interest in submitting your work to PeerJ.

With Thanks

Reviewer 3 ·

Basic reporting

The authors makes significant improvements in the manuscript after revision. I am satisfied with the modifications made.

Experimental design

Experimental design is sufficient for reproducibility.

Validity of the findings

Findings seems promising.